# NOCL: NODE-ORIENTED CONCEPTUALIZATION LLM FOR GRAPH TASKS WITHOUT MESSAGE PASSING

## ABSTRACT

Graphs are essential for modeling complex interactions across domains such as social networks, biology, and recommendation systems. Traditional Graph Neural Networks, particularly Message Passing Neural Networks (MPNNs), rely heavily on supervised learning, limiting their generalization and applicability in label-scarce scenarios. Recent self-supervised approaches still require labeled fine-tuning, limiting their effectiveness in zero-shot scenarios. Meanwhile, Large Language Models (LLMs) excel in natural language tasks but face significant challenges when applied to graphs, including preserving reasoning abilities, managing extensive token lengths from rich node attributes, and being limited to textual-attributed graphs (TAGs) and a single level task. To overcome these limitations, we propose the Node-Oriented Conceptualization LLM (NOCL), a novel framework that leverages two core techniques: 1) *node description*, which converts heterogeneous node attributes into structured natural language, extending LLM from TAGs to non-TAGs; 2) *node concept*, which encodes node descriptions into compact semantic embeddings using pretrained language models, significantly reducing token lengths by up to 93.9% compared to directly using node descriptions. Additionally, our NOCL employs *graph representation descriptors* to unify graph tasks at various levels into a shared, language-based query format, paving a new direction for Graph Foundation Models. Experimental results validate NOCL's competitive supervised performance relative to traditional MPNNs and hybrid LLM-MPNN methods and demonstrate superior generalization in zero-shot settings.

## 1 INTRODUCTION

Graphs, as versatile data structures, have become essential in modeling complex systems across diverse fields in natural and social sciences. Numerous real-world scenarios can be effectively represented as graphs, including social networks (Tang et al., 2009), brain networks (Xu et al., 2024), recommendation systems (Ying et al., 2018; Ma et al., 2019), protein interactions (Hamilton et al., 2017), fraud detection (Dou et al., 2020), and traffic networks (Wu et al., 2020; Gao et al., 2020). Message Passing Neural Networks (MPNNs) have emerged as useful tools for analyzing graph-structured data, primarily due to their ability to leverage structural dependencies and propagate information across nodes through message passing mechanisms.

However, standard MPNNs like GCN (Kipf & Welling, 2016a) and GAT (Veličković et al., 2017) typically rely heavily on supervised learning, which limits their robustness and generalization across different datasets. Each new dataset often necessitates retraining the model entirely, hindering their practical deployment. Self-Supervised Learning (SSL) approaches, such as GCA (Zhu et al., 2021), GraphMAE (Hou et al., 2022), and S2GAE (Tan et al., 2023), have emerged to enhance the generalization ability of MPNNs by pre-training models on unlabeled data through auxiliary tasks. Nevertheless, SSL methods still require fine-tuning with labeled data specific to downstream scenarios, limiting their applicability in contexts where high-quality labels are scarce or unavailable.

Recently, leveraging Large Language Models (LLMs) for graph-based applications has gained attention (Jin et al., 2024), due to their impressive generalization capabilities demonstrated in natural language processing (NLP) tasks. These models, such as GraphGPT (Tang et al., 2024), GIANT (Chien et al., 2021), and GLEM (Zhao et al., 2022), primarily focus on textual-attributed graphs

(TAGs) and node classification tasks. Other methods such as LLM-ICL (Guo et al., 2023) and LLM4Mol (Qian et al., 2023) integrate LLMs with molecule graph data by utilizing simplified molecular input line entry systems, but they neglect critical molecular node features like radical electrons and chirality, limiting their generalization. Moreover, a common challenge faced by these methods is the increased computational overhead due to longer token sequences, restricting practical applicability and scalability.

To effectively integrate LLMs with graph tasks, we identify three critical challenges:

- **C1.** Extending LLM's applicability from TAGs to non-TAGs.

- **C2.** Maximizing the utilization of raw textual node features while maintaining manageable token lengths.

- **C3.** Preserving LLM's reasoning and generalization capabilities across all graph tasks.

To address the aforementioned challenges, we propose Node-Oriented Conceptualization LLM (NOCL) with two key techniques: *node description* and *node concept*. The node description translates diverse node features into natural language paragraphs, thereby generalizing LLM's usage from TAG to non-TAG scenarios. Our node concept uses a pretrained language model (PLM) to encode node descriptions into compact embeddings, significantly reducing token lengths while preserving rich contextual information. Additionally, we introduce the graph representation descriptors to represent graph structures as textual sequences and reformulate all downstream graph tasks into human-readable queries. By transforming graph tasks into text-based comprehension problems, NOCL aligns naturally with the next-token prediction paradigm of LLMs, allowing them to directly generate task-specific outputs without relying on specialized architectural heads or rigid task-specific formats. This design not only supports standard classification tasks but also opens the door to open-ended reasoning over graphs, such as generating natural language explanations or answering free-form graph-based questions. While our current experiments focus on fixed-format tasks, this expressiveness represents a step toward more adaptive and general-purpose Graph Foundation Models (GFMs). Furthermore, by leveraging the inherent reasoning and generalization capabilities of LLMs, NOCL demonstrates strong zero-shot abilities—handling novel tasks or adapting to unseen domains without retraining. Architecturally, NOCL takes a deliberate step away from traditional MPNNs, which suffer from known limitations such as oversmoothing, locality bias, and difficulty in modeling long-range dependencies. By adopting an MPNN-free approach, NOCL offers a more flexible, scalable framework for reasoning over graph-structured data.

Our contributions can be summarized as follows:

- We introduce a novel MPNN-free method, named NOCL. With our *node description* and *node concept* components, we incorporate rich semantic node features of both TAG and non-TAG into LLMs, while significantly reducing input token lengths, facilitating model tuning on commercial-grade hardware.

- Our approach aligns LLM with graph tasks at various levels and various types of graphs uniformly, paving a new way for comprehensive GFMs without MPNNs.

- Experimental results validate NOCL competitive supervised performance relative to traditional MPNNs and hybrid LLM-MPNN methods and demonstrate superior generalization in zero-shot settings.

## 2 PRELIMINARIES

**Graph-structured data** represents information as entities (nodes) and the relationships (edges) between them. In this paper, we use $\mathcal{G} = (\mathcal{V}, \mathbf{X}, \mathbf{A})$ to denote a graph, where $\mathcal{V} = \{v_1, v_2, ..., v_n\}$ is the set of $n$ nodes. Each node $v_i \in \mathcal{V}$ is associated with a $d$-dimensional feature vector $x_i \in \mathbb{R}^d$ and $\mathbf{X} \in \mathbb{R}^{n \times d}$ denotes the node feature matrix. $\mathbf{A} \in \{0, 1\}^{n \times n}$ represents the adjacency matrix, where $a_{ij} = 1$ if there is an edge between nodes $v_i$ and $v_j$, and 0 otherwise. For node-level tasks, every node is associated with a label. For graph-level tasks, the graph is associated with a label. For edge-level tasks, every two nodes are associated with a label to indicate whether an edge exists between these two nodes.

Table 1: Comparison of capabilities of representative LLM-based graph learning models. Our NOCL offers the most comprehensive capabilities.

| Methods | Graph Type | | MPNN-Free | Tasks | | | Open-end Potential |
| --- | --- | --- | --- | --- | --- | --- | --- |
| | TAG | non-TAG | | node | link | graph | |
| LLM-GNN (Chen et al., 2023) | ✓ | | | ✓ | | | |
| TAPE (He et al., 2023) | ✓ | | | ✓ | | | |
| InstructGLM (Ye et al., 2023) | ✓ | | ✓ | ✓ | ✓ | | ✓ |
| GraphGPT (Tang et al., 2024) | ✓ | | | ✓ | ✓ | | ✓ |
| Mol-Instruction (Fang et al., 2023) | | ✓ | ✓ | | | ✓ | ✓ |
| InstructMol (Cao et al., 2023) | | ✓ | | | | ✓ | |
| LLM4Mol (Qian et al., 2023) | | ✓ | ✓ | | | ✓ | ✓ |
| ReLM (Shi et al., 2023) | | ✓ | | | | ✓ | ✓ |
| MolCA (Liu et al., 2023) | | ✓ | | | | ✓ | ✓ |
| NOCL (ours) | ✓ | ✓ | ✓ | ✓ | ✓ | ✓ | ✓ |

## 3 METHODOLOGY

### 3.1 MOTIVATION

The pursuit of a universal GFM capable of reasoning across diverse graph types and solving a wide array of graph tasks is a crucial next step in the field (Mao et al., 2024). Inspired by the success of LLMs, recent efforts have explored their use in graph learning. However, building a truly general-purpose GFM remains an open challenge. As summarized in table 1, existing LLM-based approaches often fall short by primarily focusing on TAGs and tackling individual tasks in isolation. Furthermore, many struggle with zero-shot generalization and rely on MPNNs, which have inherent limitations. To address these shortcomings and pave the way for a more versatile GFM, we propose a novel, MPNN-free model that supports a wide range of graph types and tasks within a unified framework. Our approach allows the LLM to generate task-specific outputs directly, without relying on specialized architectural heads. Beyond standard tasks, our model also opens new avenues for open-ended tasks, like reasoning and explanations—e.g., explaining why a node belongs to a particular class or which functional group contributes to a molecule's specific properties.

The overall framework of our NOCL is illustrated in fig. 1. We begin by converting original node features into node descriptions. These descriptions are then encoded into compact node concept embeddings using a PLM followed by a lightweight connector module. Next, we construct a prompt by integrating the graph representation descriptors, the encoded node concepts, and a textual description of the downstream task. This prompt is fed into the LLM, which generates the task-specific response directly through its next-token prediction capabilities.

### 3.2 NODE DESCRIPTION

The node description is a natural language representation of a node. As illustrated in fig. 2, TAGs directly utilize original texts as node descriptions. In contrast, non-TAGs, such as molecular graphs, require generating descriptive language to characterize node features. For instance, in the ogbg-molhiv dataset, each node corresponds to an atom and is associated with an 8-dimensional feature vector, capturing attributes such as atomic number, chirality, formal charge, and others. We convert each atom's original numerical features into natural language using the following template:

> This atom is [atomic name]. It has a [chirality type]. Its formal charge is [formal charge number]. The radical electrons of this atom is [number of radical electrons]...

We provides more details and examples about generating node descriptions in section A.

### 3.3 NODE CONCEPT

**Definition of node concept.** Our *node concept* refers to a fixed-size embedding vector derived from encoding a node description using a PLM, such as BERT (Devlin et al., 2019).

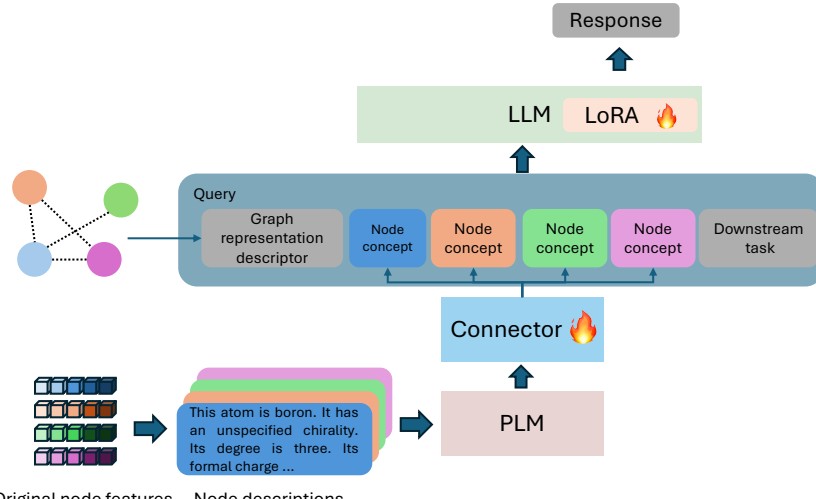

Figure 1: The overall framework of NOCL. The training process of NOCL consists of two stages. During the connector tuning stage, we freeze all parameters of the LLM and train only the connector module to align the node concept embeddings with the LLM input space. In the instruction tuning stage for downstream tasks, we freeze the connector and fine-tune the LLM using LoRA, enabling efficient adaptation to graph-related tasks with minimal additional parameters.

| | ogbn-arxiv | ogbg-molhiv |
|---|---|---|
| Graph Type | TAG | non-TAG |
| Original Node feature | $[-0.0579, -0.0525, -0.0726, …, -0.1401]_{1 \times 128}$ | $[5, 0, 3, 5, 0, 0, 1, 1, 1]$ |
| Node Description | Title: evasion attacks against machine learning at test time Abstract: In security-sensitive applications, the success of machine learning depends on a thorough vetting of their resistance to adversarial data. In one pertinent ... such systems can be easily evaded. We also sketch some countermeasures suggested by our analysis. | This atom is boron. It has an unspecified chirality. Its degree is three. Its formal charge is 5. The radical electrons of this atom is zero. Its hybridization type is SP2. It connects zero hydrogen atoms. This atom is part of an aromatic ring. This atom is part of a ring. |
| Node Concept | $[0.0013, 0.0396, -0.0254, …, 0.001]_{1 \times 768}$ | $[0.0092, -0.0887, 0.0022, …, -0.0321]_{1 \times 768}$ |

Figure 2: Node Concept Examples.

A straightforward approach to leverage node descriptions would be directly inputting them into LLMs. However, this approach faces significant limitations due to lengthy token sequences, especially prevalent in TAGs. For instance, the average node description length for the ogbn-arxiv dataset (Hu et al., 2020) is 221 tokens, requiring approximately 1 second per node for classification on an RTX 4090 GPU. Moreover, when handling graphs with numerous nodes, the aggregated token lengths quickly exceed the maximum context window of LLMs (e.g., 4096 tokens in Llama 2 (Touvron et al., 2023)).

To address these challenges, we introduce the node concept embedding, which effectively reduces node descriptions into compact embedding vectors via PLMs. Our proposed approach facilitates seamless integration of graph-structured data into multimodal LLMs (MLLMs) by:

- Requiring no architectural modifications, ensuring easy adoption and scalability.
- Significantly reducing token lengths, thus simplifying optimization and boosting efficiency.

**Node concept connector tuning.** The primary goal of node concept connector is to effectively leverage the capabilities of both the PLM and pretrain LLM. For each node description, we generate multiple single-turn conversations data, as one example illustrated as fig. 3. By treating all answers as the assistant's response and training the connector, the connector could align the embedding

> **Connector Tuning**
> Query: <|NC|>This is an embedding of an atom in a molecule. What is the element type of this atom?
> Response: This atom is boron.

Figure 3: Connector Tuning Example.

spaces of PLM and LLM together. When we train the connector, the LLM is frozen. More details and examples of connector tuning are provided in section B.

### 3.4 GRAPH TASK INSTRUCTION TUNING

**Downstream tasks reformulation** Apart from varying embedding spaces across datasets, another challenge for MPNNs is their inability to uniformly integrate graph tasks at multiple levels within a single framework. For instance, node classification tasks typically involve directly training a classifier on node embeddings. However, graph-level classification tasks necessitate additional readout functions, such as sum pooling or mean pooling (Atwood & Towsley, 2016; Simonovsky & Komodakis, 2017), to aggregate node-level information into a graph-level representation. This discrepancy constrains the performance of pre-trained models and may lead to negative transfer (Jin et al., 2020). To overcome these limitations and achieve a unified framework for all graph tasks, we adopt the prompt-learning approach proposed by Sun et al. (2023). Specifically, we reformulate node-level and edge-level tasks into equivalent graph-level tasks by constructing induced graphs centered around nodes or edges. Consequently, all tasks are effectively transformed into graph-level comprehension tasks for LLMs:1) **Node-level**: Determining the category of a node within the graph. 2) **Edge-level**: Predicting whether two nodes should be connected within the graph. 3) **Graph-level**: Classifying the category of the entire graph. This unified reformulation leverages the next-token prediction framework intrinsic to MLLMs, exploiting their powerful text-generation capabilities for graph understanding.

**Graph representation descriptors.** We represent a graph as a sequential language structure that first generates all nodes along with their corresponding node concepts, followed by the edges. Specifically, a graph $\mathcal{G}$ with $n$ nodes and $m$ edges can be described as:

$$<|BON|> \underbrace{<|NC|>_1\ 1\ \cdots\ <|NC|>_n\ n}_{2n}\ <|EON|>\ <|BOE|>\ \underbrace{<|EDGE|>\ 1\ k\cdots\ <|EDGE|>\ i\ j}_{3m}\ <|EOE|>$$

where $i, j, k \in \{1, 2, \ldots n\}$, $a_{1k} = 1$ and $a_{ij} = 1$. Special tokens introduced include <|BON|> (beginning of nodes), <|NC|> (node concept placeholder), <|EON|> (end of nodes), <|BOE|> (beginning of edges), <|EDGE|> (individual edge tuples), and <|EOE|> (end of edges). The overall token length of graph representation descriptors for $\mathcal{G}$ is $4 + 2n + 3n$. Given that multiple sequence representations can arise from varying node permutations and edge generation orders, we standardize the sequence by placing the target node as the first node for node-level tasks. For edge-level tasks, we similarly position one target node $v_i$ first and generate all nodes within its induced graph $\mathcal{G}_{v_i}$. We subsequently generate the second target node $v_j$ along with its induced graph $\mathcal{G}_{v_j}$. Edges are generated randomly within graph descriptors, except for edge-level tasks, where edges from $\mathcal{G}_{v_i}$ precede edges from $\mathcal{G}_{v_j}$.

**Instruction tuning of downstream tasks.** Leveraging the proposed graph descriptors, we construct graph instruction datasets from existing structured datasets. Each <graph, task> pair is formatted into a query-response template:

> **Query:** This is a graph: <Graph Descriptors>. <Downstream Task Query>
> **Response:** <Corresponding Text Label>

fig. 4 illustrates examples of queries and responses for various graph tasks. By utilizing pure textual responses, our NOCL approach enables seamless integration with existing MLLMs without architectural changes. We employ LoRA (Hu et al., 2022) to fine-tune LLMs on graph-structured instruction datasets, optimizing the original autoregressive training objective $\mathcal{L}_{txt}$ inherent to LLMs. To enhance the LLM's understanding of graph representation descriptors, we also incorporate node counting and edge checking problems as part of our downstream tasks. More examples of instruction tuning are provided in section C.

> **Node Classification**
> Query: This is a citation graph: <graph descriptor>. Please classify node 0 into one of the following categories: Artificial Intelligence, Hardware Architecture, ..., Systems and Control.
> Response: Information Theory
>
> **Link Prediction**
> Query: This is a citation graph: <graph descriptor>. Should node 0 connect node 7?
> Response: Yes, these two nodes should be connected.
>
> **Graph Classification**
> Query: This is a molecular graph: <graph descriptor>. Does the molecule have the ability to inhibit HIV virus replication?
> Response: Nope, it fails to exhibit any antiviral effects on HIV.

Figure 4: Graph Instruct Tuning Examples.

## 4 EXPERIMENTS

### 4.1 EXPERIMENTAL SETTINGS

**Model architectures.** For our experiments, we employ the Llama-3.2-3B-Instruct and Llama-3.2-1B-Instruct models as our base LLM and utilize the `all-mpnet-base-v2` model from Sentence-BERT (S-BERT) (Reimers & Gurevych, 2019) as the encoder for node descriptions. A simple linear layer serves as the connector bridging the node concept embeddings and the LLM.

**Datasets.** We evaluate our approach across five datasets: ogbn-arxiv (Hu et al., 2020), PubMed (Sen et al., 2008), Cora (McCallum et al., 2000), MUTAG (Debnath et al., 1991) and ogbg-molhiv (Hu et al., 2020). The detail statistics and prompts of these datasets are provided in section D. These datasets are categorized according to their respective tasks as follows: **Node level:** ogbn-arxiv, PubMed, Cora; **Link level:** PubMed, Cora; **Graph level:** MUTAG, ogbn-molhiv. We utilize all nodes from these datasets to generate the corresponding node descriptions and node concepts, which are then employed in training the connector. In our experiments, we simultaneously train models on tasks across multiple levels. For node-level tasks, we use only the training set of ogbn-arxiv. For link-level tasks, we adopt the evaluation protocol established by (Kipf & Welling, 2016b; Pan et al., 2018), partitioning the links in the Cora dataset into 85% training, 5% validation, and 10% testing subsets. An equal number of non-existing (negative) links are randomly sampled and added to each subset. For graph-level tasks, we use 80% of the MUTAG dataset and the training set of ogbg-molhiv for training purposes. Datasets not explicitly mentioned are reserved exclusively for zero-shot evaluation. We carefully ensure that there is no data leakage across the datasets utilized for different multi-level tasks.

**Metrics.** To evaluate our model's performance, we utilize two commonly used metrics: Accuracy for node classification and ROC_AUC for link prediction and graph classification.

**Baselines.** In our performance comparison, we consider various state-of-the-art methods for comprehensive evaluation. (i) MPNNs, including Graph-SAGE (Hamilton et al., 2017), GCN (Kipf & Welling, 2016a), GAT (Veličković et al., 2017), RevGNN (Li et al., 2021), GKD (Yang et al., 2022) and GLNN (Zhang et al., 2021). (ii) open LLMs, such as Baichuan-7B (Baichuan, 2023), vicunas (Chiang et al., 2023), Galactica (Taylor et al., 2022) and Llama-3.2s (Meta AI, 2024), to directly understanding TAGs. (iii) LLM-MPNNs, including: GraphGPT (Tang et al., 2024), InstructGLM (Ye et al., 2023) and InstructMol (Cao et al., 2023).

**Implementation details.** All of our models are trained using 4 NVIDIA RTX 4090 GPUs (24GB each). During the tuning phase of the node concept connector, we set the batch size to 1 per device and use a learning rate of 3e-1. For instruction tuning on downstream tasks, we use a batch size of 2 per device, a learning rate of 3e-3, and apply LoRA with a rank of 16 for fine-tuning. For inducing subgraphs in node- and link-level tasks, we use only 1-hop neighbors and ensure that the total number of nodes in the subgraph does not exceed 11. If a target node has more than 10 neighbors, we randomly sample 10 to construct the induced subgraph. To adapt LLM outputs for the ROC_AUC evaluation metric, we flatten the last hidden state corresponding to the first output token of the LLM and apply a linear projection to produce the final numeric prediction. The more details, such as other hyperparameters and training hours, are provided in section F.

Table 2: Performance comparison of various methods on node classification under both supervised and zero-shot settings. * indicates that results are obtained from the GraphGPT papers. The best results are in **bold**, and the second best are underlined.

| Methods | Training method | ogbn-arxiv supervised | PubMed zero-shot | Cora zero-shot |
|---|---|---|---|---|
| MPNNs | GraphSAGE* | 0.5480 | 0.3950 | 0.0328 |
| | GCN* | 0.5267 | 0.3940 | 0.0214 |
| | GAT* | 0.5332 | 0.3940 | 0.0167 |
| | RevGNN* | 0.5474 | 0.4440 | 0.0272 |
| | GKD* | 0.5570 | 0.3645 | 0.0470 |
| | GLNN* | 0.6088 | 0.4298 | 0.0267 |
| LLMs | Baichuan-7B* | 0.0946 | 0.4642 | 0.0405 |
| | vicuna-7B-v1.5* | 0.4962 | 0.6351 | 0.1489 |
| | Llama-3.2-1B-Instruct | 0.0549 | 0.0000 | 0.1809 |
| | Llama-3.2-3B-Instruct | 0.5141 | 0.0002 | 0.5096 |
| MPNN-LLMs | GraphGPT-vicuna-7B-1.5 | 0.6476 | **0.7011** | 0.1813 |
| | InstructGLM-Flan-T5-large | 0.7467 | - | - |
| NOCL (ours) | NOCL-Llama-3.2-1B-Instruct | 0.7440 | 0.2820 | 0.3231 |
| | NOCL-Llama-3.2-3B-Instruct | **0.7478** | 0.4764 | **0.5583** |

Table 3: Performance on link and graph tasks. * indicates that results are obtained under the zero-shot setting.

| Methods | Cora | PubMed | ogbg-molhiv | MUTAG |
|---|---|---|---|---|
| | Link | | Graph | |
| | supervised | supervised | supervised | supervised |
| GraphSAGE | 0.6789 | 0.6439 | 0.7171 | 0.6310 |
| GCN | 0.6546 | 0.6853 | 0.7256 | 0.5357 |
| GAT | 0.6591 | **0.7064** | 0.7371 | 0.5357 |
| NOCL-Llama-3.2-1B-Instruct | 0.8842 | 0.6057* | 0.7476 | **0.7262** |
| NOCL-Llama-3.2-3B-Instruct | **0.8965** | 0.6351* | 0.7576 | 0.7023 |

## 4.2 OVERALL PERFORMANCE

**Node classification.** We evaluate our approach on node classification tasks under both supervised and zero-shot settings. Results are presented in table 2. In the supervised setting, our NOCL outperforms state-of-the-art baselines. In the zero-shot setting, our proposed method achieves the best performance on the Cora dataset and ranks third on PubMed. The relatively lower performance on PubMed is likely due to the limitations of the base LLM, as both base LLMs demonstrate poor classification performance on this dataset. Nevertheless, through the integration of node concepts, graph descriptor prompts, and downstream task instruction tuning, we significantly enhance the generalization capability of our base LLMs on all datasets, including PubMed.

**Link prediction.** As shown in table 3, the proposed NOCL surpasses all MPNN baselines under the supervised setting. Remarkably, under the zero-shot setting, NOCL achieves a comparable performance to supervised GraphSAGE. Besides, MPNNs employ cosine similarity for determining edge existence, leveraging their intrinsic smoothing properties without the need for explicit link prediction. When we modify MPNNs to concatenate the outputs of target node pairs and train a classifier for link prediction, their performance drops to be around 0.55 in ROC_AUC even in supervised settings, underscoring the competitiveness of our model.

**Graph classification.** As shown in table 3, NOCL also consistently outperforms MPNNs across all evaluated datasets. Notably, on the MUTAG dataset, our approach exceeds the best-performing MPNN by a margin of 0.09 in ROC_AUC, demonstrating superior performance in graph-level tasks. Additionally, we provide a comparison for NOCL with other LLM-based methods on ogbg-molhiv dataset. As shown in table 4, our method outperforms all other methods.

Table 4: Performance on graph classification on ogbg-molhiv of LLM-based methods.

| Methods | ogbg-molhiv |
|---|---|
| Galactica-6.7B | 0.7220 |
| Vicuna-v1.3-7B | 0.5810 |
| InstructMol-G | 0.7400 |
| InstructMol-GS | 0.6890 |
| NOCL-Llama-3.2-1B-Instruct | 0.7476 |
| NOCL-Llama-3.2-3B-Instruct | **0.7576** |

Table 5: Contribution study on node classification under both supervised and zero-shot settings.

| Variants | ogbn-arxiv supervised | PubMed zero-shot |
|---|---|---|
| SD | 0.7424 | 0.2462 |
| SD+GU | 0.7291 | **0.2881** |
| NOCL (GU+Mix) | **0.7440** | 0.2820 |

### 4.3 Contribution of Graph Tasks at Various Levels

We conduct an study to investigate the contributions of different graph tasks for our proposed framework. The results are reported in table 5. All variants are based on Llama-3.2-1B-Instruct and the task is node classification. We denote: "SD": fine-tuned on a single dataset (ogbn-arxiv) only; "GU": fine-tuned with additional graph structure understanding tasks, including node counting and edge checking; "Mix": fine-tuned with a combination of graph tasks across different levels (node, edge, graph). The following observations emerge: 1) Incorporating graph structure understanding tasks improves the model's generalization ability. While the SD+GU variant performs slightly worse than SD in the supervised setting, it demonstrates significantly better zero-shot performance—indicating that structural tasks enhance the LLM's ability to generalize to unseen data. 2) Instruction mixing across task levels yields further improvements. The Mix variant, trained on a blend of node-level, edge-level, and graph-level tasks retains strong capability on single task of node classification. This suggests that NOCL benefits from multi-task instruction tuning without compromising single-task effectiveness.

### 4.4 Model Efficiency Study

Our proposed node concept significantly reduces input sequence token length during inference and training. As shown in table 6, it decreases the average sequence length from thousands to hundreds of tokens, achieving a reduction ratio of at least 81.7%. This enables highly efficient memory and time usage. For example, one data point training with a NOCL-Llama-3.2-1B-Instruct on the PubMed dataset reduces memory requirements from approximately 12.10 GB to 2.63 GB. As a result, both training and inference become feasible on commercial-grade GPUs such as the RTX 4090 (24 GB), enabling tasks, such as training on PubMed and ogbg-molhiv, that were previously infeasible due to memory constraints. Moreover, the shorter input sequences substantially accelerate training. For instance, when fine-tuning NOCL-Llama-3.2-1B-Instruct on the ogbn-arxiv dataset for one epoch, training time is reduced from 53.5 minutes (using full node descriptions) to just 8.7 minutes, achieving a 6.15× speedup.

## 5 Related Work

Recently, there has been an increasing interest in extending LLMs for graph-based applications. Depending on the role of LLMs and their interaction with MPNNs, Jin et al. (2024) have classified LLM for graphs into three categories: 1) LLM as a predictor. LLMs directly work as the final predictor for graph tasks. For instance, InstrcutGLM (Ye et al., 2023) directly converts graph structures into natural languages and apply LLMs to node classification and link prediction. 2) LLM as an encoder. LLMs extract textual features to serve as initial node feature vectors for MPNNs, which then generate node/edge representations and make predictions. These methods typically adopt an LLM-GNN cascaded architecture to obtain the final representation, such as TAPE (He et al., 2023). 3) LLM as an aligner. These methods contain an LLM component for text encoding and a GNN component for structure encoding. These two components are served equally and trained iteratively or in parallel. LLMs and MPNNs can mutually enhance each other since the LLMs can provide textual signals to MPNNs, while the MPNNs can deliver structural information to LLMs. For example, GLEM (Zhao et al., 2022) formulates the iterative training process into a pseudo-likelihood

Table 6: The statistics of average input token lengths, one-epoch training times, and GPU memory usage for training one random sampled data point across different configurations. The "OOM" indicates that the model ran out of memory during training, even with a batch size of 1. The base LLM model is `Llama-3.2-1B-Instruct`.

| Dataset | Task Level | Node Feature (tokens) | | Question Sequence (tokens) | | Training Time (min) | | GPU Memory (GB) | |
|---|---|---|---|---|---|---|---|---|---|
| ogbn-arxiv | node | 221 | | 1891 | 83.77% | 53.5 | 6.15x | 4.92 | 38.62% |
| + node concept | | 1 | | 307 | | 8.7 | | 3.02 | |
| PubMed | node | 370 | | 1731 | 90.93% | OOM | - | 7.64 | 63.31% |
| + node concept | | 1 | | 157 | | 0.68 | | 2.65 | |
| PubMed | link | 370 | | 3633 | 93.86% | OOM | - | 12.10 | 78.26% |
| + node concept | | 1 | | 223 | | 4.01 | | 2.63 | |
| ogbg-molhiv | graph | 53 | | 1639 | 81.70% | OOM | - | 6.72 | 59.23% |
| + node concept | | 1 | | 300 | | 2.05 | | 2.74 | |

variational framework, where the E-step is to optimize the LLM and the M-step is to train the GNN. Our work could be considered as using LLM as a predictor.

## 6  LIMITATION

Our approach presents several potential limitations. First, the performance of NOCL is significantly influenced by the choice of the PLM. Since our goal is to provide a new path for GFMs, we adopt the general-purpose model `all-mpnet-base-v2`; however, this model may struggle with under-represented or domain-specific language expressions. Second, although our node concept strategy enables the extension of LLM applications from TAGs to non-TAGs, generating high-quality node descriptions for non-TAGs often requires expert knowledge or auxiliary tools. Third, due to computational constraints, our experiments are conducted on only five datasets and only consider up to 10 nodes in the 1-hop induced graph. Finally, NOCL breaks the permutation invariance of MPNNs and LLMs are sensitive to token order. We provide detailed experiments addressing the third and fourth limitations in section E. It remains an open question whether the model can maintain strong generalization performance on a broader range of graph-structured data, such as those involving larger hops or more complex ordering for training. We leave this investigation for future work.

## 7  CONCLUSION

In this work, we introduced a novel LLM-centric framework that reimagines graph representation and reasoning without relying on traditional MPNN architectures. By formulating the concepts of node description and node concept, we enable LLMs to effectively operate across both TAG and non-TAG graph domains. Our design addresses three critical challenges—preserving reasoning capabilities, controlling token overhead, and extending generalizability—through a unified text-based formulation of graph tasks. This not only broadens the applicability of LLMs in graph learning but also opens a new direction toward scalable, zero-shot Graph Foundation Models (GFMs). Experimental results demonstrate that our approach achieves strong performance in both supervised and zero-shot settings, while maintaining resource efficiency. Future work will explore deeper integration of temporal and dynamic graph contexts and further optimize instruction-tuning strategies to support increasingly complex graph reasoning scenarios.

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

## 8 LLM USAGE DISCLOSURE

LLM tools (e.g., ChatGPT) were used only to assist with improving the readability and writing quality of the manuscript (e.g., grammar, phrasing, and style). No AI-generated content was used in the creation of the scientific contributions of this paper. The authors retain full responsibility for the content.

## A    MORE DETAILS ABOUT NODE DESCRIPTIONS

We provide more details about node descriptions in this section.

### A.1    TAGs

In TAGs, we directly use their raw text as node descriptions. Since, ogbn-arxiv, PubMed and Cora are citation networks and every node represent a paper. We use title and abstract of every paper as their node feature. We provide several examples as below.

**ogbn-arxiv**: We provide three examples of the node description on fig. 5.

| Original Node Feature | Node Description |
|---|---|
| $[-0.0617,$ $-0.0018$ $-0.1309,$ $-0.1314,$ $0.0524,$ $...,$ $-0.1160]_{1\times128}$ | Title: analysis and improvement of pairing free certificate less two party authenticated key agreement protocol for grid computing
Abstract: The predominant grid authentication mechanisms use public key infrastructure (PKI). Nonetheless, certificate-less public key cryptography (CL -PKC) has several advantages that seem to well align with the demands of grid computing. Security and efficiency are the main objectives of grid authentication protocols. Unfortunately, certificate-less authenticated key agreement protocols rely on the bilinear pairing, that is extremely computational expensive. In this paper, we analyze the recently secure certificateless key agreement protocols without pairing. We then propose a novel grid pairing-free certificate-less two-party authenticated key agreement (GPC-AKA) protocol, providing a more lightweight key management approach for grid users. We also show, a GPC-AKA security protocol proof using formal automated security analysis Syther tool. |
| $[0.0410,$ $0.1190$ $-0.0924,$ $-0.0839,$ $0.0801,$ $...,$ $-0.2818]_{1\times128}$ | Title: minimum area enclosing triangle with a fixed angle
Abstract: Given a set S of n points in the plane and a fixed angle 0 < omega < pi, we show how to find in O(n log n) time all triangles of minimum area with one angle omega that enclose S. We prove that in general, the solution cannot be written without cubic roots. We also prove an Omega(n log n) lower bound for this problem in the algebraic computation tree model. If the input is a convex n-gon, our algorithm takes Theta(n) time. |
| $[-0.2455,$ $-0.0077$ $-0.2020,$ $-0.1276,$ $-0.0647,$ $...,$ $0.0165]_{1\times128}$ | Title: change detection under global viewpoint uncertainty
Abstract: This paper addresses the problem of change detection from a novel perspective of long-term map learning. We are particularly interested in designing an approach that can scale to large maps and that can function under global uncertainty in the viewpoint (i.e., GPS-denied situations). Our approach, which utilizes a compact bag-of-words (BoW) scene model, makes several contributions to the problem: #R##N#1) Two kinds of prior information are extracted from the view sequence map and used for change detection. Further, we propose a novel type of prior, called motion prior, to predict the relative motions of stationary objects and anomaly ego-motion detection. The proposed prior is also useful for distinguishing stationary from non-stationary objects. #R##N#2) A small set of good reference images (e.g., 10) are efficiently retrieved from the view sequence map by employing the recently developed Bag-of-Local-Convolutional-Features (BoLCF) scene model. #R##N#3) Change detection is reformulated as a scene retrieval over these reference images to find changed objects using a novel spatial Bag-of-Words (SBoW) scene model. Evaluations conducted of individual techniques and also their combinations on a challenging dataset of highly dynamic scenes in the publicly available Malaga dataset verify their efficacy. |

Figure 5: Three examples of the node description on ogbn-arxiv dataset.

**PubMed**: We provide two examples of the node description on fig. 6.

**Cora**: We provide two examples of the node description on fig. 7.

### A.2    NON-TAGs

For non-TAGs, we need to generate descriptive language to characterize node features. Based on different original node feature, we provide different convert template as below.

| Original Node Feature | Node Description |
|---|---|
| [0.0000, 0.0000 0.0000, 0.0098, 0.0000, ..., 0.0000]$_{1\times 500}$ | Title: Social factors associated with prolonged hospitalization among diabetic children. Abstract: OBJECTIVE: To determine social factors associated with increased risk of hospital admission from diabetic ketoacidosis (DKA) or diabetic coma as well as risk of prolonged hospital stay. METHODS: A cohort of all children (</=21 years) with type 1 diabetes mellitus (DM) in the National Inpatient Sample admitted for DKA or diabetic coma during 1996 or 1997 was conducted. Patients' age, race, gender, and insurance coverage were identified. Length of stay and charges were examined; prolonged length of stay was defined as >/=7 days. RESULTS: A total of 8443 children with a primary hospital diagnosis of DKA and 123 children with type 1 DM and coma were identified; 55% of the children were girls, 32% were nonwhite, 29% received Medicaid insurance, and 33% resided in areas of poverty. Children with prolonged hospital stay were significantly more likely to be of nonwhite race (odds ratio [OR]: 2.0; 95% confidence interval [CI]: 1.6-2.5), to receive Medicaid insurance (OR: 1.4; 95% CI: 1.1-1.7), to live in areas of poverty (OR: 1.3; 95% CI: 1.1-1.7), and to be of younger age. CONCLUSIONS: When compared with state census data, nonwhite and poor children were more likely to be admitted with complications of DM and to have significantly prolonged and expensive hospital stays. These children should be targeted for intensive diabetes education and outpatient medical support both to improve their health and potentially to decrease total health care costs. |
| [0.0319, 0.0000 0.0000, 0.0165, 0.0000, ..., 0.0000]$_{1\times 500}$ | Title: Gene therapy for diabetes mellitus in rats by hepatic expression of insulin. Abstract: Type 1 diabetes mellitus is caused by severe insulin deficiency secondary to the autoimmune destruction of pancreatic beta cells. Patients need to be controlled by periodic insulin injections to prevent the development of ketoacidosis, which can be fatal. Sustained, low-level expression of the rat insulin 1 gene from the liver of severely diabetic rats was achieved by in vivo administration of a recombinant retroviral vector. Ketoacidosis was prevented and the treated animals exhibited normoglycemia during a 24-hr fast, with no evidence of hypoglycemia. Histopathological examination of the liver in the treated animals showed no apparent abnormalities. Thus, the liver is an excellent target organ for ectopic expression of the insulin gene as a potential treatment modality for type 1 diabetes mellitus by gene therapy. |

Figure 6: Two examples of the node description on PubMed dataset.

| Original Node Feature | Node Description |
|---|---|
| [0.0000, 0.0000 0.0000, 0.0000, 0.0000, ..., 0.0000]$_{1\times 1433}$ | Title: Using a Genetic Algorithm to Learn Strategies for Collision Avoidance and Local Navigation Abstract: Navigation through obstacles such as mine fields is an important capability for autonomous underwater vehicles. One way to produce robust behavior is to perform projective planning. However, real-time performance is a critical requirement in navigation. What is needed for a truly autonomous vehicle are robust reactive rules that perform well in a wide variety of situations, and that also achieve real-time performance. In this work, SAMUEL, a learning system based on genetic algorithms, is used to learn high-performance reactive strategies for navigation and collision avoidance. |
| [0.0000, 0.0000 0.0000, 1.0000, 0.0000, ..., 0.0000]$_{1\times 1433}$ | Title: Optimal Alignments in Linear Space using Automaton-derived Cost Functions (Extended Abstract) Submitted to CPM'96 Abstract: In a previous paper [SM95], we showed how finite automata could be used to define objective functions for assessing the quality of an alignment of two (or more) sequences. In this paper, we show some results of using such cost functions. We also show how to extend Hischberg's linear space algorithm [Hir75] to this setting, thus generalizing a result of Myers and Miller [MM88b]. |

Figure 7: Two examples of the node description on Cora dataset.

**ogbg-molhiv** is a molecular dataset in which each graph represents a molecule, and each node corresponds to an atom. Every node is associated with a 9-dimensional numerical feature vector. The semantic meaning of each dimension is detailed in table 7. Based on the semantics of each dimension, we convert these numerical features into natural language using a structured template:

This atom is [atomic name]. It has a [chirality type]. Its formal charge is [formal charge number]. The radical electrons of this atom is [number of radical electrons]. Its hybridization type is [hybridization type]. It connects [number of hydrogen atoms] hydrogen atoms. [This atom is part of an aromatic ring.] [This atom is part of a ring.] Its degree is [node degree].

Table 7: The meaning of every dimension on ogbg-molhiv dataset

| Index of Dimension | Meaning |
| --- | --- |
| 1 | Atomic number |
| 2 | Chirality type |
| 3 | Node degree |
| 4 | Formal charge |
| 5 | Number of connected hydrogen atoms |
| 6 | Number of radical electrons |
| 7 | Hybridization type |
| 8 | Part of an aromatic ring? |
| 9 | Part of a ring? |

**MUTAG** is also a molecular dataset. However, its node features are limited to the element type of each atom. In addition, the dataset provides edge type annotations that indicate whether a given bond is part of an aromatic ring. Based on this information, we construct natural language descriptions using the following template:

This atom is [atomic name]. [This atom is part of an aromatic ring.] Its degree is [node degree].

## B    MORE DETAILS ABOUT CONNECTOR TUNING

The goal of node concept connector tuning is to effectively bridge the representations between the PLM and pretrain LLM. We achieve this by training the LLM to reconstruct the original node description from its corresponding node concept embedding and to comprehend its high-level semantics. For each node description, we generate multiple single-turn conversation examples.

**TAGs** In TAGs, these conversations fall into three categories: 1) title recovery; 2) abstract recovery; 3) class prediction. Representative examples are provided in fig. 8. To prevent data leakage, we ensure that the class prediction task is applied only to nodes from the training set.

**Non-TAGs** For non-TAGs, we simply make LLM to reconstruct the original node description. We provide several example in fig. 9.

## C    MORE DETAILS ABOUT REFORMULATION GRAPH TASKS

We reformulate all downstream graph tasks as human-readable queries. By casting these tasks as text-based comprehension problems, NOCL aligns naturally with the next-token prediction paradigm of LLMs, enabling them to generate task-specific outputs directly—without the need for specialized output heads or rigid task-specific architectures. This design not only accommodates standard classification tasks but also enables more expressive forms of graph reasoning, such as generating natural language explanations or answering free-form, graph-grounded questions. Below, we provide detailed templates and representative examples for standard classification tasks along with their corresponding expected responses.

**Node classification** We directly prompt the LLM to predict the class of a target node from a predefined set of categories using the following template:

Please classify the node [target node index] into one of the following categories: [predefined category 1, $\cdots$, predefined category n].

---

**Title Recovery**
Query: This <|NC|> is embedding of a paper. What's the title of this paper?
Response: The title of this paper is: Spatially resolved changes in diabetic rat skeletal muscle metabolism in vivo studied by 31P-n.m.r. spectroscopy.

**Abstract Recovery**
Query: This <|NC|> is embedding of a paper. What's the abstract of this paper?
Response: The abstract of this paper is: Graphical models enhance the representational power of probability models through qualitative characterization of their properties. This also leads to greater efficiency in terms of the computational algorithms that empower such representations. The increasing complexity of these models, however, quickly renders exact probabilistic calculations infeasible. We propose a principled framework for approximating graphical models based on variational methods. We develop variational techniques from the perspective that unifies and expands their applicability to graphical models. These methods allow the (recursive) computation of upper and lower bounds on the quantities of interest. Such bounds yield considerably more information than mere approximations and provide an inherent error metric for tailoring the approximations individually to the cases considered. These desirable properties, concomitant to the variational methods, are unlikely to arise as a result of other deterministic or stochastic approximations.

**Class Prediction**
Query: This <|NC|> is embedding of a paper. What's the category of this paper?
Response: The category of this paper is: Cryptography and Security.

---

Figure 8: Examples of connector tuning for TAGs. To prevent data leakage, we ensure that the class prediction task is applied only to nodes from the training set.

---

**Node Description Recovery**
Query: This <|NC|> is embedding of an atom in a molecular. What the element type of this atom?
Response: This atom is boron.

**Node Description Recovery**
Query: This <|NC|> is embedding of an atom in a molecular. What the formal charge of this atom?
Response: The formal charge of this atom is 5.

**Node Description Recovery**
Query: This <|NC|> is embedding of an atom in a molecular. How many hydrogen atoms does this atom bond with?
Response: The atom bonds with three hydrogen atoms.

**Node Description Recovery**
Query: This <|NC|> is embedding of an atom in a molecular. How many radical electrons does this atom have?
Response: The atom bonds with zero hydrogen atoms.

**Node Description Recovery**
Query: This <|NC|> is embedding of an atom in a molecular. Is this atom part of an aromatic ring?
Response: Nope, this atom is not involved in the aromatic ring system.

**Node Description Recovery**
Query: This <|NC|> is embedding of an atom in a molecular. Does this atom belong to a ring structure?
Response: No, this atom is positioned outside of any ring structure.

---

Figure 9: Examples of connector tuning for non-TAGs

We also provide two examples in fig. 10.

**Link prediction** We prompt the LLM to determine whether two nodes should be connected using the following template:

Should node [target node index 1] connect node [target node index 2]?

The LLM is expected to respond with one of the following:

- **"Yes, these two nodes should be connected."** – indicating that an edge should exist between the two nodes.
- **"Nope, these two nodes have no relation."** – indicating that no edge should exist.

**Graph classification** We prompt the LLM to assess whether a given graph exhibits a specific property. In ogbg-molhiv,the prompt is "Does the molecule have the ability to inhibit HIV virus repli-

**Node classification**

Query: This is a graph: <Graph Descriptors>. Please classify the node 0 into one of the following categories: Artificial Intelligence; Hardware Architecture; Computational Complexity; Computational Engineering, Finance, and Science; Computational Geometry; Computation and Language; Cryptography and Security; Computer Vision and Pattern Recognition; Computers and Society; Databases; Distributed, Parallel, and Cluster Computing; Digital Libraries; Discrete Mathematics; Data Structures and Algorithms; Emerging Technologies; Formal Languages and Automata Theory; General Literature; Graphics; Computer Science and Game Theory; Human-Computer Interaction; Information Retrieval; Information Theory; Machine Learning; Logic in Computer Science; Multiagent Systems; Multimedia; Mathematical Software; Numerical Analysis; Neural and Evolutionary Computing; Networking and Internet Architecture; Other Computer Science; Operating Systems; Performance; Programming Languages; Robotics; Symbolic Computation; Sound; Software Engineering; Social and Information Networks; Systems and Control.
Response: Cryptography and Security.

**Node classification**

Query: This is a graph: <Graph Descriptors>. Please classify the node 0 into one of the following categories: Diabetes Mellitus, Experimental; Diabetes Mellitus Type 1; Diabetes Mellitus Type 2.
Response: Diabetes Mellitus, Experimental.

Figure 10: Examples of reformulation node classification task

cation?". For MUTAG, the question is "Is this molecule likely to exhibit mutagenic effects on Salmonella typhimurium?".

## D  DATASETS

- **ogbn-arxiv**: The ogbn-arxiv dataset comprises 169,343 Computer Science (CS) arXiv papers, each classified into one of the 40 categories. Each paper comes with a 128-dimensional feature vector obtained by averaging the embeddings of words in its title and abstract. The embeddings of individual words are computed by running the skip-gram model over the MAG corpus.

- **Cora**: The Cora dataset consists of 2708 scientific publications classified into one of seven classes. Each publication in the dataset is described by a 0/1-valued word vector indicating the absence/presence of the corresponding word from the dictionary. The dictionary consists of 1433 unique words.

- **PubMed**: The PubMed dataset comprises 19,717 scientific publications related to diabetes, each classified into one of three categories. The citation network includes 88651 links. Each publication is represented by a TF/IDF weighted word vector derived from a dictionary of 500 unique words.

- **ogbg-molhiv**: The ogbg-molhiv is a molecular property prediction dataset. It is adopted from the MoleculeNet. All the molecules are pre-processed using RDKit. Each graph represents a molecule, where nodes are atoms, and edges are chemical bonds. Input node features are 9-dimensional, containing atomic number and chirality, as well as other additional atom features such as formal charge and whether the atom is in the ring or not. The task is to predict the target molecular properties that whether a molecule inhibits HIV virus replication or not, as accurately as possible.

- **MUTAG**: The MUTAG dataset consists of 188 chemical compounds divided into two classes according to their mutagenic effect on a bacterium. Each graph represents a molecule, where nodes are atoms, and edges are chemical bonds. Input node features are 1 dimensional, indicating the element type of the atom.

A summary of the characteristics of the datasets is given in table 8.

## E  ORDER AND HOP EXPERIMENTS

To examine the influence of the number of hops and nodes, we employ a random walk to sample the induced graph for target nodes. As shown in table 9, without retraining

Table 8: Dataset statistics

| Dataset | Task | # Graph | # Node | # Ave. Node | # Edge | #Ave Edge | # Feature | # Class |
|---|---|---|---|---|---|---|---|---|
| ogbn-arxiv | node classification | 1 | 169343 | - | 2484941 | - | 128 | 40 |
| Cora | node classification / link prediction | 1 | 2708 | - | 13264 | - | 1433 | 7 / 2 |
| PubMed | node classification / link prediction | 1 | 19717 | - | 88651 | - | 500 | 3 / 2 |
| ogbg-molhiv | graph classification | 41127 | 1049163 | 25.51 | 2259376 | 54.94 | 9 | 2 |
| MUTAG | graph classification | 188 | 3371 | 17.93 | 7442 | 39.59 | 1 | 2 |

`NOCL-Llama-3.2-1B-Instruct`, increasing the number of nodes and hops leads to a substantial performance drop. Notably, when the number of nodes is increased to 40 within 2 hops, the accuracy decreases to 0.5470.

Table 9: Nodes and hop influence for NOCL classification accuracy on ogbn-arxiv

| #Nodes | 10 | 20 | 30 | 40 |
|---|---|---|---|---|
| 1-hop | 0.7440 | 0.7216 | 0.7081 | 0.6950 |
| 2-hop | 0.7227 | 0.7041 | 0.6293 | 0.5470 |

To investigate the influence of input token order, we evaluate three ordering strategies. The first, `Original`, follows our graph representation descriptors. The second, `Non-target Change`, preserves the position of the target node while shuffling other nodes and edges. The third, `Target Change`, alters the position of the target node along with other nodes and edges. Additionally, we introduce a variant of `NOCL-Llama-3.2-1B-Instruct`, named `non-zero`, which during training accepts target nodes not only at index 0 but also at arbitrary positions. As shown in table 10, NOCL is largely insensitive to the ordering of non-target nodes and edges but shows slight sensitivity to the target node's position. This sensitivity can be effectively mitigated by augmenting training data with target nodes appearing at different positions, enabling NOCL to maintain stable performance across order variations. These results suggest that, although NOCL departs from the permutation invariance of MPNNs, it nonetheless exhibits strong tolerance to changes in input token order.

# F   MORE DETAILS ABOUT EXPERIMENTS

To effectively reproduce our experiments, we provide the code at `https://anonymous.4open.science/r/NodeConceptLLM-1B7E`

Table 10: Rrder influence of NOCL for node classification on ogbn-arxiv

| Order type | Original | Non-target change | Target change |
|---|---|---|---|
| NOCL | 0.7440 | 0.7409 | 0.6974 |
| non-zero | 0.7395 | 0.7388 | 0.7227 |

