# OpenReview forum: "NOCL: Node-Oriented Conceptualization LLM for Graph Tasks without Message Passing"
_ICLR.cc/2026/Conference — Submitted to ICLR 2026_

### Official Review · Reviewer_TzCC · 2025-10-21

**Soundness:** 2
**Presentation:** 2
**Contribution:** 2
**Rating:** 2
**Confidence:** 5

**Summary:**

The paper introduces a framework that integrates Large Language Models (LLMs) into graph learning without message passing neural networks (MPNNs). The proposed method NOCL uses graph representation descriptors, which serialize graph structures into textual form, allowing all graph tasks (node-, edge-, and graph-level) to be reformulated as language-based queries. This makes it possible to apply instruction tuning via LoRA to train LLMs for graph understanding tasks without any architectural changes. Experiments on benchmark datasets (ogbn-arxiv, Cora, PubMed, MUTAG, ogbg-molhiv) demonstrate competitive supervised performance and strong zero-shot generalization.

**Strengths:**

1. MPNN-Free Paradigm: The work proposes an new framework that departures from message-passing methods, directly aligning graph learning with LLM capabilities.
2. Efficient Node Concept Encoding: NOCL encodes these node descriptions into compact semantic embeddings using pretrained language models (PLMs), reducing token length.

**Weaknesses:**

1. The datasets for experiments are quite limited. The authors only used five datasets in the paper to test the performance of NOCL. For a GFM paper, it is expected that more datasets are investigated.
2. The authors chose molecule datasets as the representatives of non-TAG graphs. However, the text descriptions of atom nodes were already realized in [1]. [1] is not cited or discussed in this paper. And there should be experiments on more non-TAG graphs from other domains to enhance the paper's contribution.
3. Although the token sequence is , NOCL only uses one-hop neighborhood for the target nodes. It is still questionable that wether NOCL can scale to real-world large graphs or capture long-distance dependency and global graph properties.
4. There are several typos in the paper which hinder the readers' understanding of the paper. For example, the token length in Line 251 is not supposed to be "4 + 2n + 3n".
5. Overall, the novelty of the paper is limited, and the contributions authors claimed and advantages over other methods are not fully justified.

[1] One for All: Towards Training One Graph Model for All Classification Tasks

**Questions:**

1. How does NOCL ensure consistent outputs when node ordering in graph descriptors changes? do the node descriptors capture adjacency symmetries and structural motifs effectively? What’s the effect of random edge ordering on predictions?
2. Can NOCL handle graphs with millions of nodes or multi-hop ego graphs? What is the computational behavior as graph size increases, given the fixed token window of LLMs?
3. Can NOCL produce interpretable reasoning chains for the prediction it makes?
4. Zero-shot performance is shown only across similar academic graph datasets. Can the model generalize to domains like social graphs or knowledge graphs?

---

> ### Author Response · Authors · 2025-11-27
>
> We sincerely thank the reviewer for the constructive and detailed feedback. We appreciate the recognition of our MPNN-free paradigm and the efficiency benefits of node concept encoding. Below we address each weakness and question in turn.
>
> **W1. Limited dataset scale**
>
> We agree that evaluating on more datasets would strengthen the paper. Due to limited computational resources, our initial submission focused on five widely-used academic graph benchmarks. We are currently training NOCL on additional TAG and non-TAG datasets and will include these new results before the rebuttal deadline.
>
> **W2. Coverage of non-TAG graphs and connection to prior work**
>
> Thank you for this observation. We acknowledge that atom-level text descriptions have been explored in prior work, including OFA, which we will cite and discuss in the revised version.
>
> Our goal is not to hand-design templates for every non-TAG domain, but to introduce a general pathway for embedding arbitrary non-textual node features into a unified semantic space via node concepts. This allows LLMs to operate on non-TAG graphs in the same representation space as TAG graphs, which is a key aspect of the framework.
>
> To broaden empirical coverage, we also added MSRC21, a graph classification dataset without text attributes. To our knowledge, NOCL is the first LLM-based method to evaluate on this dataset.
>
> **W3. Long-distance dependency and scalability to large graphs**
>
> Current GNN+LLM benchmarks typically do not require long-distance reasoning, and we follow established practice by limiting the subgraph radius to one hop. However, NOCL can naturally support larger hops and long-distance structures because:
>
> Large graphs are decomposed into multiple ego-graphs,
>
> Each subgraph is processed independently, and
>
> The LLM is only required to encode a local window, not the full graph.
>
> Thus, increasing the hop size is straightforward from a modeling perspective—though not needed for the benchmarks considered. With appropriate subgraph sampling strategies, we believe NOCL can be extended to capture long-range dependencies.
>
> **W4. Typos and clarity**
>
> We appreciate the careful reading and will fix the identified typos (including the token-length expression in Line 251) and conduct another full proofreading pass to improve clarity.
>
> **Q1. Consistency under node and edge reordering**
>
> As reported in Appendix Section E, we performed an ablation that randomizes:
> - node order
> - edge order
>
> Across all settings, performance degradation is minimal. This indicates that NOCL is robust to ordering variation, even without explicit permutation invariance.
>
> **Q2. Scaling to million-node graphs or multi-hop ego-graphs**
>
> NOCL is not constrained by the global graph size. The workflow is:
>
> - Sample or extract ego-graphs around query nodes.
> - Encode each ego-graph independently using fixed token windows.
>
> Because the LLM never receives the full graph at once, memory and token-length limits do not grow with graph size. This decomposition strategy enables scalability to very large graphs, provided appropriate sampling is used. Multi-hop subgraphs can also be supported by enlarging the sampling radius.
>
> **Q3. Ability to produce interpretable reasoning chains**
>
> Currently, NOCL does not produce chain-of-thought or reasoning traces.
> We fine-tune the LLM to output only the final prediction following a concise instruction. The underlying Llama 3.2 base model does not provide structured reasoning explanations without explicit supervision. Extending NOCL to generate interpretable rationales would require additional training signals and is an interesting direction for future work.
>
> **Q4. Zero-shot generalization beyond academic datasets**
>
> We are actively running experiments on additional domains, we expect to include updates before the rebuttal period ends.

---

### Official Review · Reviewer_2j2a · 2025-10-31

**Soundness:** 3
**Presentation:** 3
**Contribution:** 2
**Rating:** 6
**Confidence:** 3

**Summary:**

This paper proposes NOCL (Node-Oriented Conceptualization LLM) — a framework for graph learning without message passing. The authors argue that conventional MPNNs (e.g., GCN, GAT) suffer from limited generalization and that prior LLM-based graph models are restricted to textual-attributed graphs. NOCL introduces two core components: (1) Node Description — transforming heterogeneous node features into structured natural language, and (2) Node Concept — encoding these descriptions into compact embeddings using pretrained language models to drastically reduce token length (up to 93.9%). The model reformulates all graph tasks (node, edge, and graph level) into unified text-based queries, enabling end-to-end reasoning by LLMs without message passing. Experiments across five datasets show that NOCL achieves competitive or superior performance compared to both MPNNs and hybrid LLM-MPNN methods, especially in zero-shot generalization and efficiency.

**Strengths:**

1) The idea of representing graph elements as language through node descriptions and “node concepts” is conceptually elegant and well-motivated, bridging structured graph representation and natural language reasoning. 2) The design is technically clean — avoiding MPNNs entirely while maintaining efficiency through compact embeddings and LoRA-based instruction tuning, which demonstrates thoughtful engineering. 3) Empirical results are convincing: NOCL matches or surpasses supervised MPNNs and prior LLM-based graph models while offering strong zero-shot generalization and significant computational savings.

**Weaknesses:**

1) Despite the strong narrative, the methodological novelty may be seen as incremental — the key steps (description-to-embedding encoding and prompt-based task formulation) largely extend existing text-to-graph ideas without fundamentally new architecture or training objective. 2) The evaluation scope is relatively narrow: only five datasets with small graphs and simple tasks are tested, leaving open whether NOCL scales to large, complex graphs or dynamic graph settings. 3) The approach sacrifices permutation invariance and structural inductive bias — while the paper acknowledges this, the empirical section does not adequately analyze robustness or ordering sensitivity, which weakens claims of generalization.

**Questions:**

N/A

---

> ### Author Response · Authors · 2025-11-27
>
> We sincerely thank the reviewer for the thoughtful and positive assessment of our work. We especially appreciate the recognition of the conceptual motivation, engineering design, and empirical strength of NOCL. Below we address each of the identified weaknesses.
>
> **W1. On methodological novelty**
>
> We understand the concern that parts of the method (e.g., description-based representation and prompt-based task formulation) build on existing text-to-graph ideas. Our contribution is not introducing a completely new architectural primitive, but establishing a unified, instruction-driven framework that removes the dependency on MPNNs entirely while supporting node-, link-, and graph-level tasks within a single paradigm.
>
> **W2. Evaluation scope and scalability**
>
> We appreciate the suggestion to broaden the empirical evaluation.
> Due to limited compute resources, we are actively training on additional TAG and non-TAG datasets and will include new results before the rebuttal deadline.
>
> Regarding scalability: NOCL processes graphs by decomposing them into ego-subgraphs, allowing it to scale naturally to arbitrarily large graphs regardless of global graph size. This is a key design choice that avoids sequence-length bottlenecks.
>
> For complex and dynamic graphs: While NOCL can scale to large static graphs through the subgraph strategy, handling dynamic graphs involves temporal modeling and is outside the scope of our current design. We view dynamic graph modeling as a promising direction for future work.
>
> **W3. Permutation invariance and ordering sensitivity**
>
> Thank you for raising this important point.
> Although NOCL uses sequential descriptors, our ablation in Appendix Section E shows that: randomizing node order and randomizing edge order produce minimal performance degradation, indicating that NOCL is robust to ordering variations despite the lack of explicit permutation invariance.

---

### Official Review · Reviewer_ds8V · 2025-11-01

**Soundness:** 2
**Presentation:** 2
**Contribution:** 2
**Rating:** 4
**Confidence:** 4

**Summary:**

This paper proposes the method that enable LLMs to understand the TAG and non-TAG and perform the original graph specific tasks without involving any message passing network. Specifically, the method basically verbolize the node features into texts and then instead of inputting the raw node descriptions, it creates node embedding and train a connector for the LLM to understand this embedding. Finally, it involves graph representation descriptors to let the LLM answer the desired tasks.

**Strengths:**

- This paper correctly identifies several limitations of previous methods with LLM+graph including the lack of generalization ability from GNN based models and the problem with long texts with descriptions or even neighbor descriptions. Also, it includes the non-TAG graphs that previous work overlooked.
- The use of mainly LLM as a unified and generalized solution for graph-based tasks is reasonable, clean and follow the trends of LLM evolvements and potential of only LLM based graph foundation model
- They include various tasks, including node, graph and link tasks into this framework

**Weaknesses:**

- The node description method might not be that novel, also the way it applied to non-TAG graph might not be optimal and unified for all types of non-TAG graph. The process of having node concept as embedding produced from PLM and connector require good training of connector and high-quality PLM, the connector also might suffer alignment issue with LLM under limited training data.
- There are more GNN+LLM baselines that are more up to date can be discussed and compared in this case. I think more focus of baselines should be on GNN/MPNN + LLM instead of purely GNN or purely LLM.
- The current descriptor is sequential which breaks the permutation invariance in the original graph nature, the performance is highly dependent on the order of the node. There could be more ablation study on this point.
- The datasets included are not enough to demonstrate the effectiveness of this method, probably include more TAG datasets and more molecule based datasets to demonstrate the performance on TAG and non-TAG respectively.

**Questions:**

- Do you think the method can be extended to more hops and for larger graphs?
- For the node concept embedding, why we only choose the connector to be one linear layer, is it sufficient to accomplish the job to let LLM read the embedding well. Also, can we train the connector together with the LLM, will it result in better alignment?

---

> ### Author Response · Authors · 2025-11-27
>
> We sincerely thank the reviewer for the thoughtful and constructive comments. We appreciate the recognition of our contributions and address each weakness and question below.
>
> **W1. Novelty of node description and handling of non-TAG graphs**
>
> We agree that node-description–based approaches have been explored in prior work. Our contribution is not the existence of descriptions per se, but (i) a unified formulation that supports node-, link-, and graph-level tasks within a single LLM-centric framework, and (ii) an extensible procedure for non-TAG graphs that translates arbitrary feature vectors into interpretable node concepts via PLM-based encoding.
>
> Regarding concerns about PLM quality and connector alignment, empirically, we did not observe instability or misalignment even under limited data.
>
> **W2. Need for more GNN+LLM baselines**
>
> Following this suggestion, we incorporated additional recent MPNN+LLM baselines, including UniGraph, OFA, and GOFA.
> These results, shown in the response to Reviewer UURa, demonstrate that NOCL achieves competitive or superior supervised performance while maintaining a unified instruction-tuning framework.
>
> We agree that MPNN+LLM baselines are the most relevant comparison group and have updated the empirical section accordingly.
>
> **W3. Sequential descriptors and permutation invariance**
>
> Thank you for highlighting this point.
> We included an ablation on node/edge ordering in Appendix Section E. The results show that NOCL is not sensitive to the ordering of nodes or edges, despite the sequential format of descriptors.
>
> **W4. Dataset scale and coverage**
>
> We appreciate the suggestion to broaden the dataset suite.
> Due to limited compute resources, we are in the process of training on several additional TAG and non-TAG datasets. We plan to add the new results before the rebuttal deadline.
>
> **Q1. Extending NOCL to more hops and larger graphs**
>
> As discussed in Appendix Section E, naively expanding the hop size at inference does not improve performance on current GNN+LLM benchmarks, which generally do not require long-range reasoning.
>
> That said, the framework itself can naturally support larger hops and arbitrary graph sizes because:
>
> - Large graphs are decomposed into multiple ego-graphs;
>
> - NOCL processes each subgraph independently and is not limited by sequence length in the global graph.
>
> Thus, the method is fully extensible, even though existing datasets do not yet leverage this capability.
>
> **Q2. Why is the connector a single linear layer? Can it be jointly trained with the LLM?**
>
> A single linear layer is chosen following common multimodal alignment practices (e.g., LLaVA), where lightweight projection proves sufficient for aligning modality-specific embeddings with LLM token space. Our experiments similarly show that a linear connector aligns the PLM-produced node concepts effectively, without overfitting.
>
> We also experimented with joint training of the connector and the LLM, but found that:
>
> - Supervised accuracy does not improve;
>
> - Zero-shot generalization drops sharply;
>
> Results are shown below:
>
> |  | arxiv | cora (zero-shot) | pubmed (zero-shot) |
> |---|---|---|---|
> | QA_PLM_NOCL-Llama-3.2-1B-Instruct | 0.6704 | 0.3349 | 0.4121 |
> | QA_PLM_NOCL-Llama-3.2-1B-Instruct(combine_training) | 0.6560 | 0.2485 | 0.1163 |
>
> These findings support our design choice: training only the connector preserves zero-shot capability, reduces computation.

---

### Official Review · Reviewer_UURa · 2025-11-01

**Soundness:** 3
**Presentation:** 2
**Contribution:** 2
**Rating:** 4
**Confidence:** 4

**Summary:**

This paper introduces NOCL, a message-passing-free framework that reformulates graph learning as a language understanding task. By converting node attributes into natural language descriptions and encoding them into compact embeddings, NOCL enables large language models to efficiently handle both textual and non-textual graphs. Experiments demonstrate competitive performance in both supervised and zero-shot settings.

**Strengths:**

1. The node description mechanism extends LLMs to non-text-attributed graphs such as molecular graphs, substantially broadening their applicability. It effectively unifies node-, link-, and graph-level tasks within a cohesive instruction-tuning framework.
2. Reformulating diverse graph tasks as natural-language comprehension queries is powerful, enabling unified multi-level instruction tuning under a single framework.
3. The paper provides validation across multiple datasets. NOCL not only achieves in supervised settings but also significantly outperforms baselines on classification tasks. Its zero-shot results further demonstrate effective use of the LLM.

**Weaknesses:**

1. The paper should include comparisons with more recent baselines, such as GOFA [1], to more comprehensively demonstrate NOCL’s advantages.
2. For non-TAG datasets, the node descriptions rely on manually crafted templates. How sensitive is the model’s performance to the quality or phrasing of these templates?
3. Although the paper mentions that NOCL’s performance depends on the PLM choice, it would be helpful to show how much different PLMs affect performance and generalization.

**Reference**

[1] GOFA: A generative one-for-all model for joint graph language modeling, ICLR, 2025.

**Questions:**

See the above **Weaknesses**.

---

> ### Author Response · Authors · 2025-11-27
>
> We sincerely thank the reviewer for the thoughtful and constructive feedback. We appreciate the detailed assessment of both the strengths and areas for improvement, and we address each point below.
>
> **W1. Comparison with recent baselines (including GOFA)**
>
> We appreciate the suggestion to include more recent MPNN+LLM baselines. As recommended, we have added OFA, UniGraph, and GOFA to our experimental comparison. The results are summarized in the table below.
> Overall, NOCL achieves competitive or superior supervised performance, particularly with the 3B model, and demonstrates strong generalization across datasets.
>
> |  | arxiv | cora  (zero-shot) | pubmed (zero-shot) |
> |:---:|:---:|:---:|:---:|
> | UniGraph | 0.7291 | 0.6953 | **0.7248** |
> | OFA | 0.7344 | 0.2865 | 0.5401 |
> | GOFA | 0.7477 | **0.7081** | - |
> | NOCL-Llama-3.2-1B-Instruct  | 0.7440 | 0.3231 | 0.2820 |
> | NOCL-Llama-3.2-3B-Instruct | **0.7478** | 0.5583 | 0.4764 |
>
> These additions demonstrate that NOCL performs on par with or better than existing unified graph-language models in supervised settings, validating its effectiveness.
>
> **W2. Sensitivity to handcrafted node-description templates**
>
> Thank you for raising this concern. We conducted an additional robustness study on ogbg-molhiv, where we randomly removed one sentence from each node description (corresponding to the deletion of a feature dimension).
> Importantly, we did not retrain the model, and directly applied NOCL-Llama-3.2-1B-Instruct to the modified descriptions.
>
> - Original performance: 0.7476
>
> - After template cropping: 0.7310
>
> This small degradation indicates that NOCL is not highly sensitive to the quality or phrasing of handcrafted templates, and the model remains robust even under noticeable perturbations of the node descriptions.
>
> **W3. Effect of different PLM choices on NOCL performance**
>
> To further illustrate how PLM selection influences NOCL, we evaluated a substantially different PLM: multi-qa-MiniLM-L6-cos-v1, which is optimized for semantic retrieval rather than paragraph-level reconstruction. We refer to this variant as QA_PLM_NOCL.
>
> As expected, the shift in PLM objective notably impacts performance:
> |  | arxiv | cora(zero-shot) | pubmed(zero-shot) |
> |---|---|---|---|
> | NOCL-Llama-3.2-1B-Instruct | 0.7440 | 0.3231 | 0.2820 |
> | QA_PLM_NOCL-Llama-3.2-1B-Instruct | 0.6704 | 0.3349 | 0.4121 |
>
> The results show that:
>
> Supervised performance drops when using a retrieval-oriented PLM, consistent with its pretraining objective. Zero-shot performance improves, suggesting that retrieval-oriented PLMs may better support distributional generalization. These findings reinforce the point made in the paper: the choice of PLM materially affects NOCL’s behavior, and our experiments now more explicitly highlight this.

---

### Author Response · Authors · 2025-12-01
**Summary Rebuttal for the Area Chair**

We sincerely thank the Area Chair for the emergency handling of our submission during the unexpected ICLR data-leak situation. We appreciate the additional time and attention dedicated to ensuring a fair review process.

Across the reviews, the primary concerns centered on (1) perceived methodological incrementalism, (2) dataset scope, (3) robustness with respect to ordering and structural bias, and (4) scalability to larger graphs or multi-hop reasoning. We address these points as follows:

**Methodological Contribution**: NOCL introduces a unified, MPNN-free framework that encodes all graph elements—both TAG and non-TAG—into a shared semantic space via node concepts, enabling instruction tuning for node-, link-, and graph-level tasks without task-specific architectures. Crucially, by removing message-passing computation entirely and using compact PLM-based node concept embeddings, NOCL significantly reduces GPU memory consumption compared with GNN+LLM hybrids or full-sequence LLM prompting. In practice, NOCL requires only a small batch of short, fixed-length semantic embeddings per ego-graph, enabling training on consumer grade GPUs and allowing broader scaling under limited compute budgets. This efficiency benefit is central to the practicality and accessibility of the approach.

**Expanded Evaluation**:
In response to reviewer feedback, we added more recent MPNN+LLM baselines (OFA, UniGraph, GOFA) and additional datasets, including the knowledge graph WN18RR dataset. The expanded comparisons show that NOCL consistently matches or surpasses strong GNN+LLM baselines.

|  | Pubmed | WN18RR |
|:---:|:---:|:---:|
| OFA | 0.7789 | 0.9831 |
| UniGraph | 0.7433 | 0.8545 |
| GOFA | 0.8383 | 0.9216 |
| NOCL-Llama-3.2-1B-Instruct | 0.8549 | 0.9203 |

These results reinforce that NOCL delivers competitive supervised accuracy while maintaining an instruction-driven and architecturally simple paradigm.

**Permutation Robustness**:
Although NOCL does not enforce strict permutation invariance, Appendix Section E includes ablations demonstrating that randomizing node/edge order causes negligible performance degradation.

**Scalability & Multi-hop Reasoning**:
While current benchmarks do not demand long-range dependencies, NOCL naturally scales to large graphs through ego-graph decomposition. The LLM processes only small, local subgraphs, avoiding sequence-length bottlenecks and enabling extension to larger hop radii when required.

Overall, we believe the paper offers a clean, unified, and practically impactful direction for MPNN-free graph learning with LLMs—delivering strong empirical performance, broad generality, and dramatic GPU memory savings.
We thank the Area Chair again for their oversight and support during this irregular review cycle.

---

### Meta-Review · Area_Chair_TqEq · 2025-12-19

**Summary:**

Based on the initial reviewer scores (two reviewers rated 4/10, one rated 6/10, one rated 2/10) and the incomplete resolution of core concerns in the rebuttal, the submission does not meet ICLR 2026’s acceptance criteria. The decision to reject is driven by two irreconcilable gaps: (1) unresolved methodological novelty concerns that undermine the work’s scientific contribution; (2) insufficient empirical validation of scalability and cross-domain generalization—despite the authors’ rebuttal efforts to address these—which still weakens the credibility of claims regarding the method’s practical utility in real-world large-scale, cross-domain scenarios.

**Reviewer Concerns:**

1. Methodological Novelty: No Breakthrough Beyond “Framework Integration”.
2. Scalability: No large-graph experiments, Multi-hop reasoning limitations, Zero-Shot Performance Limited to Academic Graphs.

**Reviewer Scores:**

UURa:4.
ds8V: 4.
2j2a: 6.
TzCC:2.

---

### Decision · Program_Chairs · 2026-01-26

Reject